# Electrical Detection of Vibrations of Electrospun PVDF Membranes

**DOI:** 10.3390/ijms232214322

**Published:** 2022-11-18

**Authors:** Petr Slobodian, Robert Olejnik, Jiri Matyas, Berenika Hausnerova, Pavel Riha, Romana Danova, Dusan Kimmer

**Affiliations:** 1Centre of Polymer Systems, University Institute, Tomas Bata University in Zlin, Trida T. Bati 5678, 760 01 Zlin, Czech Republic; 2Polymer Centre, Faculty of Technology, Tomas Bata University in Zlin, Vavreckova 5669, 760 01 Zlin, Czech Republic; 3Department of Production Engineering, Faculty of Technology, Tomas Bata University in Zlin, Vavreckova 5669, 760 01 Zlin, Czech Republic; 4The Czech Academy of Sciences, Institute of Hydrodynamics, Pod Patankou 5, 166 12 Prague, Czech Republic

**Keywords:** PVDF, poly(vinylidene fluoride), electrospinning, piezoelectricity, sound sensor, electrical energy harvester, electrostriction

## Abstract

We prepared electroactive PVDF membranes, which were subjected to mechanical as well as dual electro–mechanical signals and their responses were detected by the evoked electrical pulses. The aim was to obtain primarily electric energy that could be used for light signalling, sensing of the membrane properties and membrane motion detection. The obtained data showed the unique as well as usable properties of PVDF membranes. From this point of view, the gain and analysis of the electrical responses to combined electro–mechanical loads of PVDF membranes have been important in terms of identifying the mechanism. The detected electrical response of the PVDF membrane to their electro–mechanical pulses also indicated the possibility of using this phenomenon. Among others, it suggests monitoring of membrane fouling and use for a self-cleaning mechanism.

## 1. Introduction

Transmembrane pressure, resulting in the permeation of gases and liquids through electroactive elastic membranes, as well as acoustic or electrically-driven membrane vibrations, can be used as mechanical or electrical membrane control signals. In the former case, a mechanical loading is converted to an electrical one. In the latter, an electrical loading is converted into a mechanical one. Finally, in the combined case, e.g., an electrical loading result in mechanical deformation and this induces an electrical response or vice versa. Initial mechanical and electrical loadings are used to adjust, for instance, membrane selectivity towards different chemicals, to minimize membrane fouling, to provide energy for harvesting by converting mechanical energy into electrical energy and storing it for further use. Primarily, in the mentioned applications, semicrystalline Poly (vinylidene fluoride) (PVDF), −[CH_2_−CF_2_]_𝑛_− membranes are used. The mechano–electric coupling manifests through piezoelectricity, which is the ability of PVDF to generate an electric charge in response to the deformation of covalent crystals of the *β*-PVDF crystalline face. These ferroelectric crystals, which have a zigzag/all-trans configuration, exhibit partial charges at the opposite sides of the zigzag chain [1,2], which create molecular dipolar orientation with positive hydrogen and negative fluoride leading to the spontaneous polarization of the atoms, orienting them in one direction [3]. When these covalent crystals are deformed, charge levels are shifted and electrical voltage is generated [3]. On the other hand, the electric–mechanical coupling manifests through the reverse effect of piezoelectricity, which is the ability of PVDF to generate strain and is caused by the displacement of ions in the crystal (lattice) being exposed to an electric field. Accumulation of this displacement throughout the PVDF membrane results in an overall elongation in the direction of the field.

The covalent crystals are elastic, flexible, lightweight and soluble, which makes it possible to prepare electroactive PVDF membrane networks by electrospinning [2,4]. This is an advantage of using these membranes where, on the other hand, brittle piezoelectric crystalline or ceramic materials cannot be used. Consequently, numerous applications of PVDF membranes include flexible strain sensors, biomedical sensors [4] or permeable membranes with electrically controlled selectivity towards polluting chemicals. Other cases of PVDF membrane use are actuators, wearable electronics, soft robotics [1], microelectromechanical systems, storage memory devices and energy harvesters, when the produced charge in many mechanical cycles can be store to capacitor to increase electric power for further use [5,6]. As far as the electrostriction application of PVDF membrane is concerned, a representative example is the anti-fouling membrane whereby self-cleaning is carried out by the membrane vibration using induced AC electric signals [7]. Another promising application of the piezoelectric effect is an energy harvesting.

In the present work, the prepared electroactive PVDF membranes were subjected to mechanical as well as dual electro–mechanical signals and their responses were detected by the evoked electrical pulses. The aim was to obtain primarily electric energy that could be used for light signalling, sensing of the membrane properties and membrane motion detection. The obtained data showed the unique as well as usable properties of PVDF membranes. From this point of view, the gain and analysis of the electrical responses to combined electro–mechanical loads of PVDF membranes have been the focus of our effort. The detected electrical response of the PVDF membrane to their electro–mechanical pulses indicated the possibility of using this phenomenon. Among others, it suggests monitoring of membrane fouling and use for a self-cleaning mechanism.

## 2. Results

A PVDF non-woven fibre structure was prepared by technology of electrospinning from its dimethyl formamide solution. The main idea was to prepare free standing membrane structure susceptible to mechanical deformations. PVDF fibres were pulled out from solution of PVDF powder in dimethylformamide (DMF), stretched and dried in the process of spinning and collected on polypropylene (PP) substrate. The structure of membrane was investigated by SEM analysis and results are presented in Figure 1. The upper part of the figure (a) shows a surface view of the membrane consisting of entangled PVDF fibres of different diameters and (b) a detailed view of the fibres. Porosity of PVDF membrane *ϕ* was calculated using *ϕ*_net_ = 1 − *ρ*_net_/*ρ*_PVDF_ [8], where *ρ*_net_ is the apparent density of PVDF network (*ρ*_net_ = 0.402 g/cm^3^, determined by membrane weighting and measuring dimensions) and *ρ*_PVDF_ bulk PVDF density (*ρ*_PVDF_ = 1.780 g/cm^3^). PVDF membrane has porosity 0.77%.

Membrane cross-section with a layered longitudinal fibre structure obtained by breaking it after cooling in liquid nitrogen is shown in Figure 1c. The distribution of measured diameters is presented in Figure 1d. Fibre diameters are approximately from 50 to 300 nm with average fibre diameter 182 ± 51 nm. The average fibre diameter and fibre diameter distribution of prepared PVDF membrane were determined from 100 randomly selected fibres from SEM images.

The X ray diffraction analysis (XRD) was used to confirm the presence of *β*-phase crystals in PVDF electrospun fibres. Measured XRD spectra are presented in Figure 2, Part (a). Owing to the applied high voltage during fibre spinning, providing strong electrostatic force between electrodes and the stretching polymer chains, induced diffraction lines at 2*θ* = 18.06° and 20.3°, which indicate nonpolar *α*-phase crystals and polar *β*-phase crystals, respectively [9,10]. The crystallinity *X_c_* of prepared PVDF electrospun nanofibres was calculated using
(1)$$X_{c}=\frac{\sum A_{cr}}{\sum A_{cr}+\sum A_{amr}}\times 100\;\left[\%\right]$$
where ∑*A_arm_* and ∑*A_cr_* are the total integral areas of amorphous halo and crystalline diffraction peaks, respectively [4]. According to this analysis, the crystallinity was about 50.8%.

The *α* and *β*-phase contents were determined by means of FT-IR spectra measured with a Fourier transform infrared spectrometer Figure 2b. The fraction of *β*-phase in fibres denoted *F*(*β*) was calculated using the Lambert–Beer law [4,11]
(2)$$F\left(\beta \right)=\frac{A_{\beta }}{\left(\frac{K_{\beta }}{K_{\alpha }}\right)A_{\alpha }+A_{\beta }}$$
where *A_α_* and *A_β_* represent the absorbance at 764 and 840 cm^−1^ and coefficients *K_α_* and *K_β_* represent the absorption coefficients at 764 and 840 cm^−1^ and had values 6.1 × 10^4^ cm^2^ mol^−1^ and 7.7 × 10^4^ cm^2^ mol^−1^, respectively. Portion of polar *β*-phase crystals *β* was determined using Equation (3) [12]
(3)$$\beta =F\left(\beta \right)\times X_{c\;}\left[\%\right],$$
at a value of about 85%.

Piezoelectric response of PVDF membrane to pressure pulse at the surface (see schematic illustration Figure 3a from five identical pendulum impacts (Pendulum tester for evaluating impact resistance; POLYMERTEST, Zlin, Czech Republic). During the impact, the PVDF membrane was compressed and after the pendulum rebounded, the membrane regained its original thickness after in about five ms. The idea of this setup was to modulate mechanical vibrations on PVDF piezoelectric membrane and is potential for following energy harvesting if energy is generated in many cycles.

Another experiment showed the piezoelectric response to the repeated faded pendulum impacts. The first impact cycle had full energy. The next one faded in intensity when impact was reduced by the absorbed impact energy in the previous cycle. It could be said that the first cycle had energy 100% (it means impact energy 0.5 J and velocity 2 m·s^−1^). The second impact after rebounding had 35% of the first cycle energy and so on. The third 20% and the fourth 15%, Figure 3b. As the impact energy successively decreased, the generated piezoelectric voltage decreased as well.

Figure 3c demonstrates the piezoelectric response when the anode of oscilloscope is connected to the upper side or bottom side of the membrane. It demonstrates polarity of the membrane during deformation. Without this phenomenon, any piezoelectric effect could not exist for the membrane made of randomly arranged fibres. Here the electric dipoles randomly oscillate and the total spontaneous polarization from the electric dipoles is constant and without the mechanical stimulus, no output current or voltage can be observed [13]. So, when the anode is connected to the upper side of the membrane, a positive piezoelectric signal is generated during compression (Figure 3c; blue curves). When the pendulum bounces, the fibre structure of the membrane relaxes to its original thickness and the piezoelectric signal starts to decrease and finally to be negative. When the polarity of the fabricated piezoelectric device is connected oppositely, in respect to the oscilloscope anode, the response is reversed and the electrical output curve is flipped (red curve). This “switching polarity” [14] depends on the connection of top/bottom electrodes and then in the direction of the flow of generated electrons. This radial piezoelectricity from randomly oriented electrospun PVDF nanofibres was also demonstrated in [15]. When the anode is connected to the upper electrode, electrons move to the anode and during this pressure loading, the upper surface starts to macroscopically be more positive. This can be caused by a larger deformation of the PVDF membrane in the upper layers, while the deformation of the lower layers is thus partially dampened.

As is well known, the impact loading of PVDF membranes can be used for a piezoelectric energy harvesting. We assembled an electrical circuit with Graetz bridge and a storage capacitor to show this phenomenon Figure 4a. The Graetz bridge serves to direct to convert the generated piezoelectric AC signal to a DC one which was stored in the capacitor. After usually 20 to 160 cycles of pendulum, the circuit was short circuited to measure the discharging time of stored energy or to light up three led diodes connected in series. Time-dependency of the discharge for different number of the pendulum bounces is shown in Figure 4b and lighting the diodes after 20 bounces in Figure 4c (duration of lighting is around 8 ms according to the data from Figure 4b). The increased number of harvesting cycles increased the short-circuit voltage by up to 30 V after the 160 impact cycles.

Sound waves, which produce pressure changes over the surface of the piezoelectric PVDF membrane, generate electricity that can be used for vibration sensing [16,17] to detect sound in cochlear implants [18] and abnormalities in the heart sound [19] or for sound energy harvesting [13].

In our sound detection test of the prepared PVDF membrane, the membrane was installed into the experimental setup. There were two rings between PVDF membrane, which was fixed and tensioned with screws. Sound pressure spectra were generated by speaker (Logitech, Speaker system Z 323) in a frequency range 100–3000 Hz. Generated sound was selected to be in discrete steps at 50 Hz to record the piezoelectric response as a peak-to-peak value. Figure 5a,b represents the membrane piezoelectric response to seven handclaps.

As mentioned above, the piezoelectric response of the PVDF membrane to deformation depends on its coupling to electrical polarization. However, the effect is inherently reversible. A deformation can occur as a result of the displacement of ions in the crystals being exposed to an electric field. To show it, we prepared a membrane and the experimental setup whose membrane responded with mechanical deformation to the DC input denoted as *U*_DC_. Then the inner electro–mechanical coupling induced the output voltage shown in Figure 6. If the AC voltage was applied, then the induced PVDF membrane mechanical vibrations generated a corresponding sine voltage delayed in degrees, as shown in Figure 7. 

## 3. Discussion

This study introduced electroactive PVDF membranes whose electrospun structure manifested the mechano–electric coupling through piezoelectricity and electro–mechanical coupling through inverse piezoelectricity. When the PVDF membrane was under mechanical stress, deformation of the covalent crystals of the *β*-PVDF crystalline face generated an electric charge while the displacement of ions in the crystals of the *β*-PVDF crystalline face from an external electric field resulted in overall strain. The piezoelectric responses were detectable and measurable in units of volts, Figure 3 and Figure 4. Figure 3c demonstrates the piezoelectric response when the anode of oscilloscope is connected to the upper side or bottom side of the membrane. It demonstrates polarity of the membrane during deformation. Without this phenomenon, any piezoelectric effect could not exist for the membrane made of randomly arranged fibres. Here the electric dipoles randomly oscillate and the total spontaneous polarization from the electric dipoles is constant and without the mechanical stimulus, no output current or voltage can be observed [13]. So, when the anode is connected to the upper side of the membrane, a positive piezoelectric signal is generated during compression (Figure 3c; blue curves). When the pendulum bounces, the fibre structure of the membrane relaxes to its original thickness and the piezoelectric signal starts to decrease and finally to be negative. When the polarity of the fabricated piezoelectric device is connected oppositely, in respect to the oscilloscope anode, the response is reversed and the electrical output curve is flipped (red curve). This “switching polarity” [14] depends on the connection of top/bottom electrodes and then in the direction of the flow of generated electrons. This radial piezoelectricity from randomly oriented electrospun PVDF nanofibres was also demonstrated in [14,15,20,21]. When the anode is connected to the upper electrode, electrons go to the anode and during this pressure loading, the upper surface starts macroscopically to be more positive. This can be caused by a larger deformation of the PVDF membrane in the upper layers, while the deformation of the lower layers is thus partially dampened. Finally, the charge generated in this way could be used for sufficient energy harvesting and sound detection. When collecting the energy (the energy harvesting) by means of an electronic circuit with the Graetz bridge and the storage capacitor voltage of 70 V, the collected charge was sufficient to blink three LED diodes connected in millivolts (Figure 5b). 

The PVDF membrane loaded by sound pressure spectra in the frequency range 100 Hz–3000 Hz that was generated induced an electric response in mV units (Figure 5a). Since the membrane interfered at the same time with electro–magnetic waves, both from the electrical wires in the wall of the building (50 Hz, and intensity measured by the oscilloscope about 50 mV) and the waves generated by the speaker, the experimental setup was placed into the Faraday cage made of a special steel mesh constructed for the electromagnetic shielding (electro-smog shielding metallic fabric, mesh width 1.0 mm, wire diameter 0.16 mm, mesh thickness 0.32 mm, open area 70%, weight 260 g/m^2^). The shielding was capable of suppressing the AC 50 Hz voltage to intensity below about 150 µV. The measured response to sound that was generated by clapping hands in the detected PVDF membrane response was in units of millivolts (Figure 5b).

The key results of our study were obtained as a reaction of our PVDF membranes to the electro–mechanical loading. If the AC voltage was applied and then induced, the PVDF membrane mechanical vibrations would generate the corresponding and significant sine piezoelectric voltage.

The data were collected simultaneously by oscilloscope, where from one channel of the AC generator with T-splitter for measurement of AC stimuli and from a second channel the generated piezoelectric response were connected to two different inputs of the oscilloscope. It can be expressed as a value of its amplitude and has significant signal delay as a peak shift in degrees (Figure 7). The AC stimuli of 20 V (peak-to-peak) and frequency 100 Hz generate sine mechanical deformation which induce piezoelectric voltage response with an amplitude of approximately 40 mV (peak-to-peak) and 100 Hz frequency and piezoelectric voltage with delay of 276°. In case of an input deformation, the AC voltage of 10 V (peak-to-peak) and a frequency of 600 Hz, the amplitude of the piezoelectric response was 236 mV and the time delay of the piezoelectric response was 295°. The detected electrical response of the PVDF membrane to their electro–mechanical pulses indicated the possibility of using this phenomenon. The measurements also pose that it is not only the frequency of the stimulating AC voltage moderate phase shift of the piezoelectric signal, it also has a significant effect on the generated piezoelectric voltage, measured as the peak-to-peak amplitude value. A following paper will focus on this phenomenon and our preliminary results reporting that there is a piezoelectric response to AC frequency stimuli from mHz to an approximate size of MHz with two transitions at frequencies of around 60 Hz and 90 kHz. In addition, the maximal amplitude of generated piezoelectric voltage depends on units of AC stimuli voltage. It increases linearly on the AC voltage peak-to-peak from 3, 5, 10, 15 and 20 V reaching the maxima of 85, 137, 286, 408 and 557 mV, respectively, in plateau foe between frequencies approximately 7–20 kHz. The response of the substances to the action of an electric field has two limiting cases. It is an irreversible change, a time-dependent transport of electrically charged particles that are present in the substance. An electric current is created, and substances are called electrical conductors. Alternatively, electrical displacement occurs due to the formation of induced dipoles, or dipoles already present in the substance, such as in PVDF covalent β crystals as C-F bonds (with partial charge δ−) and C-H bonds (δ+). This accumulation of energy is then reversible and disappears after the field has been extinguished. The electrical shift is generally time-dependent, and a phase shift is induced between the input stimulus and the response [22,23,24].

A future work will focus on the detection of the filtration process of aqueous solutions and dispersions (variables can be like change of solutions viscosity, electrical conductivity, filtration of dyes or steroids or particles such as kaolin and algae). Detection of filter fouling will be based both on the amplitude of the generated piezoelectric voltage and by changing the frequency spectrum of the AC stimulated membrane. Subsequently, experiments about the possibility of using electrostriction to clean the filter will start.

## 4. Materials and Methods

### 4.1. Electrospinning of PVDF Membrane

PVDF polymer was purchased from Arkema as a brand Kynar Flex^®^ 2801, Zhejiang, China. It had a powder state and was dissolved in dimethylformamide (DMF) from Brenntag CR s.r.o, Chropyne, Czech Republic. Both were of analytical grade and no additional purification was necessary. PVDF in DMF was prepared as 20 wt. % PVDF solution by dissolving 2 g of polymer in 8 g DMF as solvent. The solution was then stirred with a magnetic stirrer at 340 rpm at 40° C for 2 hrs. The process of electrospinning was performed at room temperature (22–24 °C) and a relative humidity about 31–34%. The construction of electrospinning apparatus (Spin line 40, Zlin, Czech Republic, Figure 1) was carried out with 32 static hollow stainless steel, each having an inner diameter of 1.5 mm. The voltage between needles and collecting electrode was 65 kV, at a distance of 20 cm and a dosing volume of 2.3 cm^3^ per hour. Nanofibre networks were collected at two different times of 10 and 30 min of electrospinning process with a final thicknesses of 18 and 400 µm, respectively. PVDF nanofibres were collected during processing on spun-bond prepared non-woven polypropylene, collected in the capacity of 30 g/m^2^.

### 4.2. Analysis

The structure of prepared PVDF network was analyzed using the scanning electron microscope—SEM (NOVA NanoSEM 450, FEI Co., Hillsboro, OR, USA) at an accelerating voltage of 10 kV. The average nanofibre diameter and fibre diameter distribution were analyzed from the obtained SEM pictures using Digimizer, version 5.4.4, MedCalc Software, Ostend, Belgium and one hundred measurements of randomly selected parts of fibres.

The crystalline content was calculated from data obtained by *X*-ray diffraction—XRD (MiniFlex^TM^ diffractometer, Malvern Panalytical, Malvern, UK) with *CoK_β_* radiation operated at 40 kV and 15 mA. The samples were scanned in a 2*θ* range of 10–90°. The collected data using cobalt source were converted to copper source using the program 15 PowDLL software [25,26].

Fourier transform infrared (FT-IR) spectrometer (NICOLET iS5, Fisher, Thermo Scientific, Waltham, MA, USA) was used to check the content of *β* phase fraction in the prepared PVDF fibres. The apparatus worked in attenuated total reflection (ATR) mode using a diamond crystal in the wavenumber range of 400–4000 cm^−1^ at a resolution of resolution 4 cm^−1^.

### 4.3. Piezoelectric and Electrostriction Measurements

Several measurements to prove and quantify piezoelectric and electrostriction performers for the prepared PVDF polymer membrane structure composed of electrospun fibres were performed. An oscilloscope (Infinivision 1000 x-series, 4ch, 100 MHz, DSOX1204A, Keysight, Santa Rosa, CA, USA) was used to measure the generated piezoelectric voltage. In this course, three types of experimental setups with incorporated PVDF electrospun non-woven membrane were prepared, as presented in Figure 2. Part (a) of the schematic illustration represents the experimental setup for mechanical loading on the top surface of the PVDF membrane by pressure pulses using an instrument for the impact flexibility of rubbers (Polymer test, Versta, Zlin, Czech Republic). There are two cupper electrodes to measure the generated piezoelectric potential, the electrodes made of Cuprexite (double layer structure containing Cu foil fixed by glass fibre epoxy plate) between which PVDF membrane is fixed with a thickness of 400 µm. Finally, a steel plate (18 × 18 mm) was placed on top of the membrane to avoid its destruction in the area of the pendulum impact. The energy of impact was 0.5 J with an impact velocity of 2 m/s.

Figure 2a shows the experimental setup for the detection of sound. Sound pressure spectra were generated by speaker (Speaker system Z 323, Logitech, Newark, CA, USA) in a frequency range 100–3000 Hz. Generated sound was selected to be in discrete steps 50 Hz to record piezoelectric response of mV as a peak-to peak value from a distance of 20 mm and a sound intensity of 100 dB (Figure 5a). In Figure 5b, the plot shows the electrical response to clapping hands 30 cm from the top surface of fixed and tightened PVDF membrane. The diameter of the PVDF membrane was 50 mm with a thickness of 18 µm. The membrane was fixed with the help of plastic three-dimensional (3D) printed acrylonitrile butadiene styrene (ABS) parts. The sound of clapping creates sound pressure waves that vibrate the stretched membrane, which generate an electric voltage detected and measured with help of an oscilloscope.

Figure 2c represents the experimental setup for proving and demonstrating the electro–mechanical response. There are two electrical circuits. The first causes electrostrictive stimuli by direct current (DC) of 30 V with the help of power supply Metex AX 502 (AEMC Instruments, Dover, NH, USA). When the power supply was turned on, it led to electrostriction of PVDF membrane (30 × 30 mm and thickness 400 µm). The deformation of PVDF membrane by DC stimuli was measured by the perpendicular pair of electrodes creating the second electric circuit as the piezoelectric response following deformation by electrostriction.

## 5. Conclusions

A PVDF membrane was prepared by the technology of electrospinning from polymer solution. When fibres were solidified under tension, it led to the formation of the molecular dipolar orientation of polymer chains with zigzag/all-trans configuration. The representative data of membrane ability to generate electricity in response to mechanical deformation as well as to electro–mechanical deformation have been shown. In particular, the detected electrical response of the PVDF membrane to their electro–mechanical pulses suggested the possibility of using this phenomenon. Among others, e.g., for monitoring of membrane fouling and use for a self-cleaning mechanism. For instance, induced membrane mechanical vibrations can prevent particles from fouling the membrane and thus cleaning it.

## Data Availability

The data presented in this study are available on request from the corresponding author, P.S.

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
