# Peer review of "Electrical Detection of Vibrations of Electrospun PVDF Membranes"

_ijms, 2022, doi:10.3390/ijms232214322_

Round 1

Reviewer 1 Report

This paper reported PVDF-based piezoelectric devices for energy harvesting. As far as the reviewer knows, similar research work has been widely studied and reported, and the author has not proposed to make innovations in materials, structures, properties and applications. Therefore, the reviewer doesn't think this article has much value in publishing in this journal. The author must put forward the novelty and highlights of this research work.

1. The full text does not seem to mention the application of piezoelectric devices in pressure sensors. Please carefully check the title and text.

2. The main problem of electrospun electronic devices is the stability of long work cycle. Please analyze the working stability of reported piezoelectric devices.

Author Response

Dear Reviewer.

I'm sorry to say that due to the short deadline for the required extensive correction of the first version of our paper, we preferred to completely rewrite
 that version. In the new version of the paper, we emphasized new original data on the electrical response of the PVDF membrane to electro-mechanical
 stimulation by AC. This reaction has not yet been measured and published.  At the same time, it promises many practical applications,
 such for a self-cleaning of filtering membranes, monitoring the flow through the membranes, etc.
We would be very happy if this new version could be published in your International Journal of Molecular Sciences.

Sincerely

Petr Slobodian

Centre of Polymer Systems, University Institute, Tomas Bata University in Zlin, Trida T. Bati 5678, 76001 Zlin, Czech Republic.

Reviewer 2 Report

The article used three types of experimental setups to test PVDF electrospun membranes. They were capable of generating electrical voltage in response to pressure stimuli containing pressure pulse or sound waves. The generated charge can be harvested by a capacitor which can light up three LED diodes connected in series. For electrostriction, electrostrictive stimuli reached the maximum at around 77 mV. The results are interesting. However, it is not novel in the field of piezoelectric PVDF. Only three devices were reported, however, their logic relationship or potential theoretical is not analysed. It is more like an experimental report rather than an academic article. It should be improved. I recommend it can be published in another journal. To improve the manuscript, I still have some suggestions as below.

1. In the abstract, the author should highlight the novelty and the importance of the work.

2. For line 122-143, line 148-164 and line 166-182, the three paragraphs are repeated totally.

3. For line 177, did the 12 V come from Fig. 3b? The peak-to-peak value may be around 14 V. The author should give an average value with an error bar.

4. For line 315, it is better to give which kind of pressure pulse was used. The details should be given.

5. For line 286, Schematic illustration 1 is shown behind Schematic illustration 2. 

6. What is the relationship between Fig.5 a and Fig.5 b? It seems like only the frequency spectra were shown and it has no relationship with Fig.5 b. If the author can find out which frequencies are sensitive to the PVDF membrane, it may be more interesting.

Author Response

(The authors gave the same response as above.)

Reviewer 3 Report

Reviewer's  comments:

Reviewer #: In the present work entitled " Poly (vinylidine fluoride) electrospun membrane with piezoelectric properties applied as pressure sensors, energy harvesting and electrostriction," the authors have fabricated and investigated PVDF membrane using electrospinning technique for multiple applications likewise pressure sensor, energy harvesting, and electrostriction. They have presented their results satisfactorily with adequate explanations. However, a few corrections should be made for further consideration. This contribution gives new information; it is well-written.

Before publication, the authors must correct the mistakes in the text that are found all over the manuscript and address the following:

1.      A better characterization of the materials and technologies used to make the membranes is required.

2.      A better characterization of the piezoelectric properties as a pressure sensor, energy harvesting, and electrostriction used in research is required.

3.      Each keyword should start with a capital letter.

4.      The graphical abstract is too complex to present. Please draw a new figure which can generalize the theme with a concise design.

5.      The Introduction should consist of five paragraphs answering the following five questions: What is the problem? Why is it interesting and important? Why is it hard? Why hasn't it been solved before? (or, what's wrong with previously proposed solutions? What are the key components of my approach and results?

6.      It is not clear what is new in this study. The novelty of the presented work should be highlighted in a proper way. as other researchers have already conducted the same work and characterization of the same materials?

7.      The author should mention the sensitivity of the prepared membrane, which is missing in the manuscript.

8.      The reviewer did not find the reply to the following questions: (i) What value does the paper add? (ii) What is the purpose of the paper? It is not clear what new knowledge we will gain after reading this work. Corresponding statements should be added to the manuscript in regard to the achievements in this research field. Corresponding statements should be added to the manuscript in regard to the achievements in this research field. Moreover, the Authors should exhibit the knowledge gap which the present study fills.

9.      It is necessary for sustainability analysis of these types of membranes (economic, social, environmental)

10.  Comparison between obtained results and literature data is a very weakness of this work. Besides, the obtained results are not compared with published data by other researchers. For more contribution, the Authors should compare their results with those in relevant published works of other researchers. If it is possible, The Authors should compare their results with those of previous works.

11.  Mention how mechanical properties are measured, particularly which ASTM standard is adopted here?

12.  The author should explain how much beta percentages have been increased from their original state. In contrast, PVDF polymer performance would be improved if incorporated into nanomaterials.

13.  The author should explain which factor or parameter strategy increases the beta of the PVDF membrane, as beta is important for piezoelectric pressure sensors.

14.  The author should also explain in detail, without incorporating nanomaterials and parallel or alignment of nanofibers, how the author got good beta and piezoelectric performance and energy harvesting properties.

15.  The author should need to add at least 3-5 new references from anywhere or especially from the "International Journal of Molecular Sciences" relevant to his topics; you can add these references to your manuscript which are relevant to your work and helpful, https://doi.org/10.1007/s10854-019-01751-w &  https://doi.org/10.1007/s10854-021-07590-y

16.  The author should explain how they measure the porosity of the materials.

17.  Conclusions should be more concrete, and future research directions presented

18.  Overall, the work is interesting; it needs to follow the suggestions to improve the manuscript.

Author Response

(The authors gave the same response as above.)

Round 2

Reviewer 2 Report

Although the authors have modified the manuscript. The following issues should be clearly addressed before it can be accepted.

1. The author tried to show how much modification they did, and provided a really complex marked version. And the complex, marked version increased the workload and difficulty for reviewers. I suggest both the marked version and the clean version should be uploaded.

And if the author changed anything according to the reviewers’ suggestions. The changes should be shown one by one, letting the reviewers know how and where the changes are.

2. Still for the marked version, I cannot calculate the length or the word count. I am not mean the longer the better, but I think the description of the results is too simple and less. The authors did three different types of experiments in this work, it should have many things to describe.

3. Page 6, line 244. The author said, “the membrane has regained its original thickness after 6 ms”. However, in the new version of figure 3a, only 5 ms of the figure was given. The 6 ms even longer time period figure should be given.

4. Page 6, second paragraph

(a) “It could be said that the first cycle had energy 100%. The second impact after rebounding had 35 % of the first cycle energy and so on. The third 20 % and the fourth 15 %, Fig. 3b. As the impact energy successively decreased, the generated piezoelectric voltage decreased as well.” Where is the energy value calculated from? Was it calculated from the piezoelectric voltage from Figure 3b or from the repeated faded pendulum? If it was calculated from the repeated faded pendulum, could the author please give the relationship between the energy and the piezoelectric voltage?

(b) And I also wonder why the author used the pendulum. In most published piezoelectric-related papers, they usually used a motor to provide different forces and frequencies on the piezoelectric materials to prove their piezoelectric property. What’s the reason or advantage of choosing the repeated faded pendulum? Or it has any relationship with the purpose of the manuscript?

(c) The description for Figure 3c is better to follow the description of Figure 3b. Same reason for Figure 5 b. It makes readers confused.

5.  Page 6, line 284, how long will the diodes be lighting?

6.  For Figure 4, the signal after the Graetz bridge can also be recorded and added to Figure 4.

7.  Figure 5b, what’s the meaning of “one section of amplitude 2 mv”? Are there any grids there? It is hard to see and know how big the signal is.

8.  For Figure 7

(a) How could the author be sure that the time shift was 7.5 ms and 1.21 ms, not the 7.5+T ms and 1.21+T ms?

(b) Could the author keep the frequency and only change the input AC voltage? Could the author keep the input AC voltage and only change the frequency? If possible, the author could give more figures related to different frequencies (same input voltage) and different input AC input voltages (same frequency)? And also can give a new relationship of shift time and voltage with the frequency and input voltage.

9. It is hard to say which experiments will be done in the next, future work. As the author mentioned, membrane fouling may be related to this paper. Could the author try to collect any air pollution using the PVDF membrane, such as for automobile exhaust? After a different collecting time, the membrane can be measured using the devices in the manuscript.

And I didn’t see any relationship in your work related to self-cleaning. Could you please explain?

Author Response

Dear reviewer,                                                                                            November 2, 2022

   We are sending you a revised version of our Manuscript ID: ijms-1931629. It was originally entitled ”Poly (vinylidine fluoride) electrospun membrane with piezoelectric
properties applied as pressure sensors, energy harvesting and electrostriction”. Now, because of major revision with a new name ”Electrical Detection of Vibrations of Electrospun PVDF Membranes” by Petr Slobodian, Robert Olejnik, Jiri Matyas, Berenika Hausnerova, Pavel Riha, Romana Danova and Dusan Kimmer, for a possible publication in International Journal of Material Science - Frontiers in Polymeric Materials.

First of all, we have to thank to you for many inspiring comments. The revised version of the paper takes into accounts all of them. Using the “Track Changes” (by red/yellow color highlighting), the changes are easily visible. The detailed replies are below. 

Once again thank you very much for your help in the review process.

Yours sincerely,      P. Slobodian

Comment 1

I suggest both the marked version and the clean version should be uploaded.

Reply: Clean version can be easily change from marked version. Please, just to make letter only black and no yellow background.

Comment 2.

Still for the marked version, I cannot calculate the length or the word count. I am not mean the longer the better, but I think the description of the results is too simple and less. The authors did three different types of experiments in this work, it should have many things to describe

Reply: Description of the results was expanded as much as possible.

Comment 3

  1. Page 6, line 244. The author said, “the membrane has regained its original thickness after 6 ms”. However, in the new version of figure 3a, only 5 ms of the figure was given. The 6 ms even longer time period figure should be given.

Reply: It was corrected.

  1. Page 6, second paragraph

(a) “It could be said that the first cycle had energy 100%. The second impact after rebounding had 35 % of the first cycle energy and so on. The third 20 % and the fourth 15 %, Fig. 3b. As the impact energy successively decreased, the generated piezoelectric voltage decreased as well.” Where is the energy value calculated from? Was it calculated from the piezoelectric voltage from Figure 3b or from the repeated faded pendulum? If it was calculated from the repeated faded pendulum, could the author please give the relationship between the energy and the piezoelectric voltage?

Reply: Piezoelectric response of PVDF membrane to pressure pulse at the surface (see schematic illustration 2a) from five identical pendulum impacts, (Pendulum tester for evaluating impact resistance; POLYMERTEST Zlin, Czech Republic.). During the impact, the PVDF membrane was compressed and after the pendulum rebounded, the membrane has regained its original thickness after in about five ms. The idea of this setup was to modulate mechanical vibrations on PVDF piezoelectric membrane and is potential for following energy harvesting if generated energy in many cycles. Another experiment showed the piezoelectric response to the repeated faded pendulum impacts. The first impact cycle had full energy. The next one faded in intensity when impact was reduced by the absorbed impact energy in the previous cycle. It could be said that the first cycle had energy 100% (it means impact energy 0.5 J and velocity 2 m.s-1). The second impact after rebounding had 35 % of the first cycle energy and so on. The third 20 % and the fourth 15 %, Fig. 2b. As the impact energy successively decreased, the generated piezoelectric voltage decreased as well.

(b) And I also wonder why the author used the pendulum. In most published piezoelectric-related papers, they usually used a motor to provide different forces and frequencies on the piezoelectric materials to prove their piezoelectric property. What’s the reason or advantage of choosing the repeated faded pendulum? Or it has any relationship with the purpose of the manuscript?

Reply: The idea of this setup was to modulate mechanical vibrations on PVDF piezoelectric membrane and is potential for following energy harvesting if generated energy in many cycles.

(c) The description for Figure 3c is better to follow the description of Figure 3b. Same reason for Figure 5 b. It makes readers confused.

Reply: It was corrected.

  1. Page 6, line 284, how long will the diodes be lighting?

Reply: Fig. 4c (duration of lighting is around 8 ms according data from Fig. 4b).

  1. For Figure 4, the signal after the Graetz bridge can also be recorded and added to Figure 4.

Reply: It is demonstrated by Fig. 4 part b) as a discharge of stored voltage for different number of impact cycles.

  1. Figure 5b, what’s the meaning of “one section of amplitude 2 mv”? Are there any grids there? It is hard to see and know how big the signal is.

Reply: Figure 5. a) Frequency spectra of generated piezoelectric voltage by a PVDF membrane as a response to sound generated by the speaker in a frequency range 100 - 3000 Hz, b) membrane piezoelectric response to seven handclaps. (Note: one section is represented by the range between two grid lines on the graph background represented by thin white lines).

  1. For Figure 7

Reply: There significant delay between stimulated AC signal and piezoelectric response indicated in figures in units of degree.

(a) How could the author be sure that the time shift was 7.5 ms and 1.21 ms, not the 7.5+T ms and 1.21+T ms?

Reply: The data were collected simultaneously by oscilloscope where from one channel of AC generator with T-splitter one for measurement of AC stimuli and second for generated piezoelectric response were connected to two different inputs of the oscilloscope.

(b) Could the author keep the frequency and only change the input AC voltage? Could the author keep the input AC voltage and only change the frequency? If possible, the author could give more figures related to different frequencies (same input voltage) and different input AC input voltages (same frequency)? And also can give a new relationship of shift time and voltage with the frequency and input voltage.

Reply: These measurements are part of the newly prepared manuscript. These dependencies are very interesting, but we still lack their interpretation. Therefore, we apologize that we cannot publish them fully yet. To the current manuscript we added this statements:

A next paper will focus on this phenomenon and our preliminary results poses that there is piezoelectric response to AC frequencies stimuli from mHz to approximately units of MHz with two transitions at frequencies at around 60 Hz and 90 kHz. In addition, maximal amplitude of generated piezoelectric voltage depends on size of AC stimuli voltage. It increase linearly on AC voltage peak-to-peak from 3, 5, 10, 15 and 20 V reaching maxima as 85, 137, 286, 408 and 557 mV in plateau for each between frequencies approximately 7-20 kHz.

  1. It is hard to say which experiments will be done in the next, future work. As the author mentioned, membrane fouling may be related to this paper. Could the author try to collect any air pollution using the PVDF membrane, such as for automobile exhaust? After a different collecting time, the membrane can be measured using the devices in the manuscript.

Reply: It was partially answered in previous Reply. However, we also added this:

A future work will focus on the detection of filtration process of aqueous solutions and dispersions (variables can be like change of solutions viscosity, electrical conductivity, filtration of dyes or steroids or particles like kaolin, algae etc.). Detection of filter fouling will be based both on the amplitude of the generated piezoelectric voltage and by changing the frequency spectrum of the AC stimulated membrane. Subsequently, experiments about the possibility of using electrostriction to clean the filter will start.

Needless to say, thanks to your well-founded comments we were able to modify our manuscript considerably. Thank you very much indeed.

Yours sincerely,             P. Slobodian

Reviewer 3 Report

After thoroughly reviewing the author's revised manuscript (ijms-1931629-peer-review-v2), I conclude that he has provided his answers to the reviewers' suggested comments with appropriate explanations.

Therefore, I accept and recommend this version related to the International Journal of Molecular Sciences. 

Author Response

                                                                                                   November 1, 2022

Dear reviewer,

   We are sending a revised version of our Manuscript ID: ijms-1931629. It was originally entitled ”Poly (vinylidine fluoride) electrospun membrane with piezoelectricproperties applied as pressure sensors, energy harvesting and electrostriction” by Petr Slobodian, Robert Olejnik, Jiri Matyas, Berenika Hausnerova, Pavel Riha, Romana Danova and Dusan Kimmer.

Once again thank you very much for your help in the review process. In addition, thank you for accepting our paper for publication.

Yours sincerely,      P. Slobodian

Round 3

Reviewer 2 Report

No more comments.